# Effectiveness of transcranial direct current stimulation on hand dexterity in stroke patients: a protocol for a systematic review and meta-analysis

Lulwa Alabdulaali ,[1,2] Lydia Hickman,[3] T David Punt,[1] Ned Jenkinson[1,4]

[1]School of Sport, Exercise and Rehabilitation Sciences, University of Birmingham, Birmingham, UK
[2]Department of Physical Therapy, College of Applied Medical Sciences, Imam Abdulrahman Bin Faisal University, Dammam, Saudi Arabia
[3]School of Psychology, University of Birmingham, Birmingham, UK
[4]Centre for Human Brain Health, University of Birmingham, Birmingham, UK

**Correspondence to**
Mrs Lulwa Alabdulaali;
laa886@student.bham.ac.uk

## ABSTRACT

**Introduction** Hand dexterity is the ability to execute the skilful movements using the hand and fingers. It is commonly impaired poststroke resulting in a profound deterioration in the quality of life for patients with stroke. Transcranial direct current stimulation (tDCS) is a form of non-invasive brain stimulation, which has gained a popularity as an adjunct therapy in recovering motor dysfunction poststroke. Promising results have been gained from applying tDCS in combination with motor rehabilitation, however, the outcome of tDCS on the upper limb motor function poststroke has been varied. Different results are potentially related to the discrepancy of the area of brain stimulation. Therefore, we aim to enhance the application of tDCS to improve its effectiveness in recovering hand dexterity through testing our hypothesis that stimulating the primary motor cortex could improve fine dexterity more than gross dexterity.

**Methods and analysis** This protocol has been reported according to Preferred Reporting Items for Systematic Review and Meta-Analyses Protocols guidelines. CENTRAL, MEDLINE, EMBASE, SCOPUS, Web of Science and CINAHL databases will be searched with no restriction in language and publication date. The selected studies will be randomised controlled trial investigating the effect of tDCS alone or in combination with motor rehabilitation in improving hand dexterity of patients with stroke with upper limb hemiparesis. The outcomes of interest are fine and gross hand dexterity measures. Two independent reviewers will assess the eligibility of the study, extract data and appraise the methodological quality. The data will be pooled in a meta-analysis if applicable or interpreted narratively. Grading of Recommendations, Assessment, Development and Evaluation approach will be used to assess the overall quality of evidence for the fine and gross dexterity measures.

**Ethics and dissemination** Ethical approval is not required for this study. The dissemination plan is to publish the results in a peer-review journal and presenting results in a conference.

**PROSPERO registration number** CRD42021262186.

### Strengths and limitations of this study

► This study will include the three stages of recovery of patients with stroke.
► The search strategy designed to access both published and non-published studies.
► The search strategy is not limited to publication date or language.
► This study will be limited to the randomised controlled trials.

limb hemiparesis is muscle weakness and poor dexterity on the side of the body opposite of the side of brain lesion.[3 4] Dexterity refers to 'the fine, voluntary movements used to manipulate small objects during a specific task, as measured by the time required to complete the task' (p. 209).[5] There are two types of dexterity: fine and gross. Fine dexterity is the ability to perform a delicate manipulation of very small object by precise movements of the fingertips.[6] Gross dexterity is the ability to make a manipulation of large object by coordinated arm-hand movement.[6] The majority of daily activities, such as self-care, dressing and eating, require a combination of fine and gross dexterity in combination with other upper limb movements.[7] Therefore, maintaining hand dexterity is an important determinant of quality of life for patients with stroke.[8]

However, patients with stroke with upper limb hemiparesis demonstrate dexterity dysfunction. They perform movements in a slow, uncoordinated manner because their hand movements lack accuracy and proper finger sequencing.[3 9] Likely more than half of the patients with stroke have persistent hand dexterity deficiency at 6 months poststroke. Kwakkel *et al* conducted a prospective cohort study,[2] in which they recruited 102 patients with stroke from seven different hospitals. All the patients were enrolled in an

## INTRODUCTION

Stroke is a common cause of disability in the UK.[1] Upper limb hemiparesis is a common impairment poststroke, with 76% of patients with stroke suffering from upper limb hemiparesis.[2] The clinical presentation of upper

intensive rehabilitation programme, with the study procedure starting within 2 weeks of stroke occurrence. Periodic follow-up assessments of the upper and lower limbs function were conducted. For hand dexterity they used the Action Research Arm Test (ARAT), which is a reliable and valid tool for assessing hand function.[10 11] At first assessment, no dexterous hand movement was reported in the ARAT. At 6 months, 38% showed improvement in functional ability, while 62% persisted with loss of hand dexterity.[2]

Stroke rehabilitation involves assessing and improving hand dexterity in patients with stroke in order to improve upper limb function.[12] Many functional assessment tools have been established to measure the level of upper limb motor function. These tools include tests for muscle strength and gross and fine hand dexterity. Examples of such tests include the Fugl Meyer Test,[13] Wolf Motor Function Test[14] and ARAT.[10] There has also been considerable interest in examining the efficacy of different techniques to enhance upper limb function poststroke, with some positive results described with a variety of methods as repetitive task training,[15] constraint-induced movement therapy[16] and non-invasive brain stimulation.[17 18]

A recent development has been the use of transcranial direct current stimulation (tDCS), a form of non-invasive brain stimulation. tDCS device includes a portable electrical current generator and two electrodes: cathode and anode, and works by delivering a low intensity constant electrical current to the brain that alters the cortical excitability.[19] It is a neuromodulatory technique that has gained popularity for use in recovering motor dysfunction and cognitive impairment.[12] It has been used to improve motor function among patients with stroke, patients with Parkinson's disease and old people.[17 20–22] Increased brain plasticity is thought to be the underlying mechanism of functional improvement.[19] Due to tDCS' promising effectiveness, cost-effectiveness, ease of application, safety and tolerability, recent years have seen increased investigation of use.[12 19 23 24]

Studies exploring the effect of stroke on the brain using functional MRI have observed decreased neuronal activity of the ipsilesional side few days poststroke.[25] This reduced activity in turn decreases interhemispheric inhibition from the ipsilesional to the contralesional side, leading to an increase in contralesional neuronal activity, which increases interhemispheric inhibition from the contralesional to the ipsilesional side. This abnormal increase of interhemispheric inhibition likely restricts voluntary movement and therefore adversely affects motor recovery.[24–26] Given these abnormal patterns of activity, tDCS has been used in an attempt to produce focal changes in neuronal activity to rebalance brain activity between the two hemispheres. The focal changes are either depolarising (diminishing the contralesional excitability), hyperpolarising (increasing the ipsilesional excitability) or both, depending on the tDCS montage.[27 28]

Researchers have investigated factors to enhance the application of tDCS to increase its effectiveness, for example, the area of brain stimulation and the orientation of the electrodes.[19] There are three montages of tDCS application described in the literature in improving upper limb motor recovery poststroke. The first montage is anodal tDCS. In this montage, the anodal electrode that produces a polarising effect is placed on the ipsilesional primary motor cortex (PMC), and the cathode electrode is used as a return electrode to close the electric circuit. It is placed on the contralesional supraorbital region.[12 19] In the cathodal tDCS montage, the cathode electrode that produces the depolarising effect is placed on the contralesional PMC. The anode electrode serves as the return electrode and is placed on the ipsilesional supraorbital region.[12 19] In the bilateral tDCS montage, the anodal electrode is placed over the ipsilesional PMC and the cathode is placed over the contralateral PMC.[12] Moreover, in controlled studies, tDCS can be applied as a sham stimulation. The sham tDCS is the method to blind the control group. The participant receives a very weak and short duration stimulating current, at a level where it could hardly produce any stimulation.[29]

tDCS has shown potential effectiveness in improving upper limb function poststroke when conducted both alone[22] or in combination with motor rehabilitation.[18 30] Several randomised controlled trials (RCTs) investigated the augmentation of the rehabilitation plasticity effect through conjunct therapy.[18 30 31] Conjunct therapy involves tDCS and motor rehabilitation. These studies found a superior beneficial effect of conjunct therapy in upper limb recovery compared with rehabilitation only.

## RATIONALE

Given the essential role of the upper limbs in daily life, the recovery of upper limb motor function poststroke is an important issue for stroke rehabilitation community.[12] The efficacy of tDCS in improving hand function has been shown to vary between studies.[20] Discrepancies in results could be related to heterogeneity in the application of tDCS, for example, the brain stimulating area.[12 32] The study done by Hummel et al[17] investigated the effect of stimulating the primary motor cortex (PMC) using tDCS on upper limb function, using the Jebsen-Taylor Hand Function Test as an outcome measure. It includes subtests for both fine and gross hand dexterity. Hummel et al found that the fine movements, like picking up paper clips, are more improved than gross movement like moving cans. This might be because the distal muscles of the upper limb receive higher input from the PMC (through the corticospinal tract) than the proximal muscles.[33]

Moreover, work from our laboratory[34 35] has shown a preferential effect of tDCS stimulating the PMC on movements of the hand as opposed to movements of the whole arm among healthy young and older adults. Given the PMC is the target of choice in studies investigating tDCS in upper limb recovery in stroke, this raises the question of whether a similar preferential recovery of fine versus gross hand dexterity is seen in these studies.

We will conduct this systematic review and meta-analysis to enhance the tDCS application, therefore optimising its therapeutic benefit in upper limb motor recovery post-stroke. We will throughly investigate if there are differences in the recovery between fine and gross hand tasks. Specifically, we will investigate our hypothesis that stimulating the PMC using tDCS could improve the fine hand dexterity more than the gross dexterity because the fine hand tasks mainly require fingertip grasping while the gross hand tasks involve a combination of proximal arm movement with hand grasping.[6]

## OBJECTIVES

This research aims to answer our research question "what is the difference in the tDCS effect between fine and gross hand dexterity of stroke patients with upper limb hemiparesis?"

Specific objectives are:
1. To evaluate the effectiveness of tDCS on fine hand dexterity of stroke patients with upper limb hemiparesis.
2. To evaluate the effectiveness of tDCS on gross hand dexterity of stroke patients with upper limb hemiparesis.
3. To compare the effect of tDCS between fine and gross hand dexterity.
4. To investigate the impact of patient characteristics on the effect of tDCS.

## METHODS AND ANALYSIS

This protocol has been reported according to Preferred Reporting Items for Systematic Review and Meta-Analyses (PRISMA) Protocols guidelines.[36]

### Eligibility criteria
#### Types of studies
*Inclusion criteria*
► Randomised controlled double-blinded or single-blinded trials. Using either parallel or crossover design.
► English and non-English studies.
► Full-text studies.
► Published and unpublished studies.

*Exclusion criteria*
► Non-randomised clinical trial.
► Conference abstract.

#### Types of participants
The participants needed to be:
► Patients with stroke aged 18 years and above, both male and female patients with upper limb hemiparesis.
► Diagnosed with ischaemic or haemorrhagic stroke, either cortical or subcortical.
► In the acute (1–7 days poststroke) or subacute (7 days to 6 months poststroke) or chronic phase (6 months or more poststroke) when the intervention is applied.[37]

► Suffering from mild or moderate or severe upper limb impairment. The cut-off points of Fugl Meyer Assessment of the upper extremity will be used to classify the impairment into: severe (score 0–28), moderate (score 29–42) and mild (score 43–66).[38]

### Types of intervention
#### Intervention group
The participants have received one or multiple sessions of tDCS that conducted alone or with motor rehabilitation. The tDCS either applied before or simultaneously with motor rehabilitation. The tDCS stimulated the primary motor cortex through one of the following montages: anodal, cathodal and bilateral. The motor rehabilitation includes any technique of motor training for upper limb.

#### Control group
The participants have received one or multiple sessions of sham tDCS or motor rehabilitation or sham tDCS plus motor rehabilitation.

### Types of outcome measures
The outcomes of interest are as follows:
► Validated functional measures for fine hand dexterity; assess the ability to perform a delicate manipulation of very small object by precise movements of the fingertips.[6] For instance, Purdue Pegboard Test and Nine Hole Peg Test.
► Validated functional measures for gross hand dexterity; evaluates the ability to make a manipulation of large object by coordinated arm-hand movement.[6] For example, Box and Blocks Test and Minnesota Manual Dexterity Test.

### Information sources
The study is planned to start on 1 August 2021 and the anticipated completion date of the study is 30 December 2021.
► We will conduct database and non-database searches with no restriction on publication date.
1. Database search
   The following databases will be searched:
   CENTRAL (Cochrane Central Register of Controlled Trials).
   MEDLINE (Ovid).
   EMBASE (Ovid).
   SCOPUS.
   Web of Science.
   CINAHL (Cumulated Index of Nursing and Allied Health Literature).
2. Non-database search
   We will review the reference list of the included studies.
► We will conduct grey literature search including ProQuest Dissertations and Theses, Open Access Theses and Dissertations and Electronic Thesis Online Service databases.

## Search strategy
An example of search strategy for searching MEDLINE is available as an online supplemental file 1.

## Study records
### Data management
The search records from the selected databases including the study title, abstract will be exported to a reference management software (EndNote X9). The EndNote X9 will organise the records and recognise the duplicate studies.

### Selection process
After removing the duplicate reports using EndNote X9, two independent reviewers will assess the eligibility of the study and discard the irrelevant ones. Initially, the two reviewers will review the studies title and abstract according to the inclusion criteria and exclude the irrelevant studies. Then the full text will be obtained and screened for those studies that potentially meet the inclusion criteria and if more details are required to confirm study exclusion. The two reviewers will decide about the study inclusion and will make discussion if there is disagreement. The third reviewer will make the final decision if the two reviewers cannot reach an agreement. The PRISMA flow diagram will be used to present records of the selection process with clarifying the reason of exclusion.[39]

## Data collection process
### Data extraction
Two independent reviewers will review the full text of the included studies and extract the required data using a custom data extraction sheet. The third reviewer will be consulted if there is any disagreement between the two reviewers.

### Dealing with missing data
The corresponding author of the included studies will be contacted if there is missing information.

### Data items
The two reviewers will extract the following data from the included studies:
1. Bibliographic details: study title, authors and year.
2. Study characteristics: design, level of blinding and participants allocation.
3. Participants characteristics: sample size, gender, age mean, handedness, stroke type, stroke location, time since stroke onset and severity of upper limb impairment.
4. The intervention group: tDCS montage, electrode size, intensity, density and duration, the technique of motor rehabilitation if conducted and the number of treatment sessions.
5. The control group: the type of intervention: sham tDCS plus motor rehabilitation or motor rehabilitation only.
6. Outcome measures: the fine dexterity measures and gross dexterity measures.
7. Results: the result of outcome of interest (improve, no change or deteriorate).

## Outcome and prioritisation
The primary outcomes of interest are fine dexterity measures and gross dexterity measures.

## Risk of bias in individual studies
To discover any possibility of risk of bias within the included studies, the two reviewers will critically assess the methodological quality of the included studies using the Physiotherapy Evidence Database (PEDro) scale. The PEDro scale is a valid and reliable tool in critically appraising the RCTs.[40 41] It comprises 11 items and each one is scored either 1 if present or 0 if not. The total score is out of 10 because the first item is not included in the total score. After appraising the included studies, we will use the cut-off points defined by Foley et al,[42] to classify each study according to PEDro score into one of the following: excellent quality (score 9–10), good quality (score 6–8), fair quality (score 4–5) and poor quality (score below 4).

## Synthesis of results
The fine and gross dexterity measures will be analysed as continuous variables. The analysis will be based on within-subject comparisons for both parallel and cross-over design RCTs. We will obtain the sample size, mean and SD for the baseline and postintervention scores of the fine and gross dexterity measures for both the intervention and control groups from each included study. We will contact the corresponding authors if there is missing data.

### Data analysis
All data analysis will be conducted using the R software. First, we will quantify the Cohen's d effect size for the difference between baseline and the postintervention score of the fine and gross dexterity measures for the intervention and control groups from each included study. Second, we will calculate pooled effect sizes and CIs of various groups of studies via meta-analyses in R software. We will display separate tables for each meta-analysis containing the corresponding sample sizes and effect sizes of the studies included in the sample.

### Assessment of heterogeneity
We will use the $I^2$ test to assess the heterogeneity statistically.[43] The $I^2$ is a common test being used in meta-analysis. It measures the variability percentage in the effect sizes caused by the variability between studies rather than sampling error.[43] Its score will be between 0% and 100%, with 0% reflecting no heterogeneity and 100% indicating very high heterogeneity.[44]

### Meta-analysis
Meta-analysis will be performed using the fixed-effects model if the studies are similar ($I^2 < 25\%$).[45] If there is

heterogeneity ($I^2 > 25\%$),[45] the random-effects model will be used. We will run a total of four main meta-analyses:

1. Fine dexterity meta-analysis (intervention group).
2. Fine dexterity meta-analysis (control group).
3. Gross dexterity meta-analysis (intervention group).
4. Gross dexterity meta-analysis (control group).

To make comparisons between the outcomes of each meta-analysis, we will inspect the pooled effects sizes and identify whether there are overlapping CIs. If the CIs overlap, we will conclude that there is no difference between the pooled effect sizes. However, if the CIs do not overlap, we will conclude that there is a significant difference between the pooled effect sizes.

First, we will assess the treatment effect of the tDCS on fine dexterity through comparing the pooled effect sizes of fine dexterity meta-analysis (meta-analysis 1) and control of fine dexterity meta-analysis (meta-analysis 2). Second, we will evaluate the treatment effect of the tDCS on gross dexterity through comparing the pooled effect sizes of gross dexterity meta-analysis (meta-analysis 3) and control of gross dexterity meta-analysis (meta-analysis 4). Third, we will compare the effect of tDCS between fine and gross dexterity through comparing the pooled effect sizes of fine dexterity meta-analysis (meta-analysis 1) and gross dexterity meta-analysis (meta-analysis 3).

Additionally, we will run exploratory subgroup analyses to investigate the impact of patient characteristics on the effect of the tDCS. Specifically, we will split the participants according to severity of upper limb impairment (mild and moderate vs severe), stroke phase (acute and subacute vs chronic) and stroke area (cortical vs subcortical) and run separate meta-analyses for each group. Then we will compare the pooled effect sizes.

In case of considerable heterogeneity ($I^2 > 75\%$),[45] we will run subgroup analyses using the random-effects model to reveal which factors underlie the variation between the studies. If the considerable variation between studies is persistent, the data will not be pooled in a meta-analysis. The result of the individual study will be interpreted narratively and presented in a summary table.

## Meta-bias

### Assessment of publication bias

Publication bias is the potential bias of publishing studies with positive results (rejecting the null hypothesis) rather than negative results,[46] causing a biased overestimate of the intervention effect. We will assess the possibility of publication bias in our study by creating funnel plots using R software. Funnel plots are scatter plot showing the SE (y-axis) against the effect size (x-axis) of the included studies. We will visually appraise the funnel plots to detect asymmetry, which implies the possibility of publication bias. Additionally, we will conduct Egger's test in R software to quantitatively assess asymmetry.[47]

## Confidence in cumulative evidence

The overall quality of evidence of outcomes of interest of this study (fine and gross dexterity) will be judged by Grading of Recommendations, Assessment, Development and Evaluation (GRADE) approach.[48] The GRADE is a valid approach that produces implications of the evidence by evaluating the body of evidence at the outcome level rather than the study level. The assessment will be conducted separately for the fine dexterity and gross dexterity outcome measures across the following domains: risk of bias, publication bias and results consistency, directness and precision. Based on the overall GRADE score, the quality of evidence will be rating one of the four levels: high (score ≥4), moderate (score=3), low (score=2) and very low (score ≤1).[48]

## DISCUSSION

Several systematic reviews have been conducted to assess the effect of tDCS on upper limb motor function.[20 27 49–52] These previous reviews reviewed the impact of tDCS on improving upper limb functional outcomes. However, they considered the improvement in the upper limb, with no differentiation between fine and gross hand dexterity. As such, this review and meta-analysis will look for the difference between tDCS effect on fine and gross hand dexterity. Therefore, while numerous tools are available for assessing upper limb function, the outcomes of this study will be limited to reliable and valid fine and gross dexterity measures. Because several of the available tools cannot be considered a test of fine or gross dexterity only, they include subtests that assess different aspects of upper limb motor function like joints movements, muscle strength and hand dexterity without a split into fine and gross dexterity subtests.

The current study will include crossover design RCTs. A common concern with crossover trials is the carryover effect that occurs when the treatment effect of the first period of the study carries over to the next period, which could affect the participant's response.[53] While the washout period between the two treatment periods could reduce the carryover effect, the crossover design might be inappropriate to investigate the intervention with long-lasting effect.[53] However, the effect of a single session of the tDCS persists for several minutes up to 1 hour after the stimulation.[19 54] Therefore, the crossover design is widely used in tDCS studies because the possibility of the carryover effect is very low.

## ETHICS AND DISSEMINATION

Ethical approval is not required for this study. The dissemination plan is to publish the results in a peer-review journal and presenting results in a conference.

**Contributors** The protocol was substantially designed by LA and refined by NJ, LH and TDP. LH provided substantial guidance on analyses. LA drafted the manuscript. All authors edited the manuscript and read and approved the final version.

**Funding** No funds were received in support of this work. This study is a part of Lulwa Alabdulaali PhD, which is supported by a scholarship from Imam Abdulrahman Bin Faisal University, Saudi Arabia.

**Competing interests** None declared.

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
