## [Reviewer comments · BMJ Open]

ARTICLE DETAILS

TITLE (PROVISIONAL)	The effectiveness of transcranial direct current stimulation on hand dexterity in stroke patients: a protocol for a systematic review and meta-analysis.
AUTHORS	Alabdulaali, Lulwa; Hickman, Lydia; Punt, T. David; Jenkinson, Ned

VERSION 1 – REVIEW

REVIEWER	Kang, Nyeonju Incheon National University
REVIEW RETURNED	23-Aug-2021

GENERAL COMMENTS	The current protocol of the systematic and meta-analysis aims to investigate potential effects of tDCS on either fine or gross motor functions in patients with stroke. Although the topic of this meta-analysis seems to be very interesting and useful for stroke rehabilitation literature, there are some suggestions that should be considered for increasing the clarity of this study. 1. In the Introduction, the authors need to add the rationale for examining potential different effects of tDCS on fine and gross motor functions post stroke. For example, do the specific tDCS protocols (e.g., intensity less than 2 mA or timing of tDCS) reveal different levels of positive treatment effects on fine and gross motor dexterity? Why is this meta-analytic topic important for stroke motor rehabilitation?2. For characterizing patients with stroke, why do the author exclude acute stroke patients (less than 2 weeks)? Further, the chronic phase after stroke is normally defined with the time since stroke more than 6 months. Please add relevant references for the recovery phases.3. In the Methods, the definitions for the types of outcome measures look vague. Please add references and more potential outcome measures for fine and gross motor dexterity, respectively.4. For the effect size calculations, the authors plan to collect mean and SD values suitable for RCTs with between-subject comparisons. However, given that this meta-analysis may include crossover design studies, this approach cannot be used for estimating effect sizes with within-subject comparisons.5. Similarly, the authors state that they will evaluate the difference between baseline and the post-intervention across fine and gross dexterity outcome measures via two separate meta-analyses. However, it is possible that the included studies may not directly report differences in fine and gross dexterity outcome measures for tDCS and sham groups. In many cases, studies reported only values at baseline and posttest. Thus, the authors may mention various methods for calculating effect sizes.
--

	6. For the subgroup analysis, please add the specific measures for determining mild and moderate and severe motor impairments. Perhaps, applying meta-regression analysis is more useful for this research question.
--	--

REVIEWER	Rahnama, Leila University of Social Welfare and Rehabilitation Sciences,
REVIEW RETURNED	12-Sep-2021

GENERAL COMMENTS	This study entitled “The effectiveness of transcranial direct current stimulation on hand dexterity in stroke patients: a protocol for a systematic review and meta-analysis.” aimed to systematically review the existing literature on the effects of tDCS on hand dexterity in patients with stroke. This is an interesting study and would be of interest to readers. However, some concerns need to be addressed before publication. 1- Introduction:  - Introduction, in my opinion, is too lengthy. There is a lack of coherence observed in this section. Sometimes it is even more of a discussion like last paragraph page 7 and first paragraph page 8. An introduction should be concise and focused on the main aim of the study while, giving the needed background information that readers may need. - The question of the study is mentioned neither in the introduction nor in the method sections. - There is another electrode montage for tDCS named “ Unihemispheric Concurrent Dual-Site tDCS. It is a type of bilateral montage, although it is unihemispheric. It is mostly used in studies on chronic pain, but it was used to evaluate the effect of interventions on motor function in stroke patients too. Authors may find related information in the following article: - Toluee et al. The Effect of Unihemispheric Concurrent dual-site Transcranial Direct Current Stimulation of Primary Motor and Dorsolateral Prefrontal Cortices on Motor Function in Patients With Sub-Acute Stroke. Frontiers in human neuroscience, 2018. 1- Methods:  - The specific dates for the study duration were not provided. - The limitation on publication date, if any, was not provided. - There are several tests with which hand dexterity is assessed. Are BBT and PPT only outcomes of interest for this study? Is there a particular reason not to include other tests such as Box and Block and the Assembly? - I do suggest adding CINAHL to the search databases. I personally, sometimes found a paper there that I haven't found in other databases. - How about gray literature like dissertation thesis? - The authors mentioned they aimed to compare the effects of tDCS on fine and gross dexterity. But they didn't provide the information on how they would do that. - Search Syntax or search keywords was not listed. 2- Discussion:  - Protocol of systematic review should consist of a discussion section in which the prospective limitation is discussed. This article is missing that.
---

VERSION 1 – AUTHOR RESPONSE

REVIEWER #1

1) In the Introduction, the authors need to add the rationale for examining potential different effects of tDCS on fine and gross motor functions post stroke. For example, do the specific tDCS protocols (e.g., intensity less than 2 mA or timing of tDCS) reveal different levels of positive treatment effects on fine and gross motor dexterity? Why is this meta-analytic topic important for stroke motor rehabilitation?

We thank the reviewer for highlighting this point. We have rewritten the rationale to justify our hypothesis and show the importance of the study.

2) For characterizing patients with stroke, why do the author exclude acute stroke patients (less than 2 weeks)? Further, the chronic phase after stroke is normally defined with the time since stroke more than 6 months. Please add relevant references for the recovery phases.

We thank the reviewer for this point. We have now amended our eligibility criteria to include the acute phase of stroke, defined the stroke phases and added a reference.

3) In the Methods, the definitions for the types of outcome measures look vague. Please add references and more potential outcome measures for fine and gross motor dexterity, respectively. We thank the reviewer for highlighting this point. We have added definitions and specific examples of the outcome measures to the 'outcome measures' section of the methods. Also, we have written a paragraph in the discussion clarifying our perspective in including outcome measures in the study.

4) For the effect size calculations, the authors plan to collect mean and SD values suitable for RCTs with between-subject comparisons. However, given that this meta-analysis may include crossover design studies, this approach cannot be used for estimating effect sizes with within-subject comparisons.

We thank the reviewer for their concern. The analysis will be based on within-subject comparisons for both parallel and crossover design RCTs.

We will quantify the Cohen's d effect size for the difference between baseline and the post-intervention score of the fine and gross dexterity measures for the intervention and control groups from each included study.

We have now added this sentence to 'synthesis of results' section "The analysis will be based on within-subject comparison for both parallel and crossover design RCTs".

5) Similarly, the authors state that they will evaluate the difference between baseline and the post-intervention across fine and gross dexterity outcome measures via two separate meta-analyses. However, it is possible that the included studies may not directly report differences in fine and gross dexterity outcome measures for tDCS and sham groups. In many cases, studies reported only values at baseline and posttest. Thus, the authors may mention various methods for calculating effect sizes. Thank you for allowing us to clarify this point. We will not take the difference between fine and gross from the study. We will obtain the sample size, mean and standard deviation for the baseline and post-intervention scores of the fine and gross dexterity measures for both the intervention and control groups from each included study. We will then quantify the effect size for the difference between baseline and the post-intervention score. Then we will assess the effect of tDCS by identifying whether there are overlapping confidence intervals between intervention and control group pooled effect sizes for both fine and gross dexterity. We have expanded in the meta-analysis section to make this clear.

6) For the subgroup analysis, please add the specific measures for determining mild and moderate and severe motor impairments. Perhaps, applying meta-regression analysis is more useful for this research question

We thank the reviewer for the suggestion. We will use the cut-off points of Fugl Meyer Assessment of the upper extremity to classify the upper limb impairment into mild, moderate, and severe. Then, we will run meta-analysis for mild and moderate and meta-analysis for severe then we will compare the pooled effect sizes.

We have now added a sentence to 'types of participants' section of the methods showing how we will classify the upper limb impairment.

REVIEWER #2

Introduction:

1) Introduction, in my opinion, is too lengthy. There is a lack of coherence observed in this section. Sometimes it is even more of a discussion like last paragraph page 7 and first paragraph page 8. An introduction should be concise and focused on the main aim of the study while, giving the needed background information that readers may need.

We thank the reviewer for this point, but it is not clear which paragraphs the reviewer means because pages 7 and 8 are the methods section not introduction.

We think the reviewer means the three paragraphs before rationale 'paragraphs 7,8 and 9'. We have now made amendments in the introduction to make it concise and focused. We have removed part of paragraph 7 to the rationale. We have deleted part of paragraph 8 and moved last part of it to the rationale. And we have deleted paragraph 9. Accordingly, we have deleted a sentence from the abstract. Also, we have rewritten the rationale (response to reviewer #1 comment #1).

2) The question of the study is mentioned neither in the introduction nor in the method sections. We Thank the reviewer for mention this point. We have now added the research question to the 'objectives' section.

3) There is another electrode montage for tDCS named " Unihemispheric Concurrent Dual-Site tDCS. It is a type of bilateral montage, although it is unihemispheric. It is mostly used in studies on chronic pain, but it was used to evaluate the effect of interventions on motor function in stroke patients too. Authors may find related information in the following article: Toluee et al. The Effect of Unihemispheric Concurrent dual-site Transcranial Direct Current Stimulation of Primary Motor and Dorsolateral Prefrontal Cortices on Motor Function in Patients With Sub-Acute Stroke. *Frontiers in human neuroscience*, 2018.

We thank the reviewer for the valuable information. We will consider this type in including studies in our review.

Methods:

1) The specific dates for the study duration were not provided.

We thank the reviewer for this point. We have now added the planned start and end dates for the study to the start of the 'Information sources' section of the methods.

2) The limitation on publication date, if any, was not provided.

The search strategy will be conducted with no restriction on publication date. This point is mentioned at the 'Information sources' section of the methods highlighted in yellow.

3) There are several tests with which hand dexterity is assessed. Are BBT and PPT only outcomes of interest for this study? Is there a particular reason not to include other tests such as Box and Block and the Assembly?

We thank the reviewer for allowing us to clarify this point. The BBT and PPT are examples of the outcomes of interest of this study. The BBT is specific for gross dexterity. The PPT includes subtests and all of them are specific for fine dexterity. We have now added more examples to the 'outcome measures' section of the methods. And we have written a paragraph in the discussion clarifying our perspective in including outcome measures in the study.

4) I do suggest adding CINAHL to the search databases. I personally, sometimes found a paper there that I haven't found in other databases.

We thank the reviewer for the suggestion. We have now added CINAHL to the database search under 'information sources' section of the methods and in the abstract.

5) How about gray literature like dissertation thesis?

We thank the reviewer for this point. We have added databases to search gray literature to 'information sources' section of the methods. Also, we have adjusted the 'inclusion criteria' to include unpublished studies to avoid publication bias.

6) The authors mentioned they aimed to compare the effects of tDCS on fine and gross dexterity. But they didn't provide the information on how they would do that.

We thank the reviewer for highlighting this point. We will compare the effect of tDCS on fine and gross dexterity by conducting two separate meta-analyses, one for fine dexterity and the other for gross dexterity, then comparing the pooled effect sizes to identify any overlapping confidence intervals. We have expanded in the meta-analysis section to make this clear.

7) Search Syntax or search keywords was not listed.

Database search strategy for MEDLINE was attached as supplementary file 1. This is mentioned at 'search strategy' section of the methods highlighted in yellow.

Discussion:

1) Protocol of systematic review should consist of a discussion section in which the prospective limitation is discussed. This article is missing that.

We Thank the reviewer for this point. A discussion paragraph has been added.

VERSION 2 – REVIEW

REVIEWER	Kang, Nyeonju Incheon National University
REVIEW RETURNED	12-Nov-2021

GENERAL COMMENTS	The authors successfully revised the manuscript consistent with my suggestions. Now, the manuscript is suitable for the publication.
--

REVIEWER	Rahnama, Leila University of Social Welfare and Rehabilitation Sciences,
REVIEW RETURNED	29-Nov-2021

GENERAL COMMENTS	The authors have responded to all my questions and made the necessary changes to the manuscript.
--